# Leaching Characteristics of Low Concentration Rare Earth Elements in Korean (Samcheok) CFBC Bottom Ash Samples

**Lai Quang Tuan** [1,2]**, Thriveni Thenepalli** [3] **, Ramakrishna Chilakala** [4]**, Hong Ha Thi Vu** [3]**, Ji Whan Ahn** [3] **and Jeongyun Kim** [3,]*****

[1] Department of Resources Recycling, University of Science & Technology, 217 Gajeong-ro, Gajeong-dong, Yuseong-gu, Daejeon 34113, Korea; tuanlai@ust.ac.kr

[2] Tectonic and Geomorphology Department, Vietnam Institute of Geoscience and Mineral Resources (VIGMR), 67 Chienthang Street, Hadong district, Hanoi 151170, Vietnam

[3] Center for Carbon Mineralization, Mineral Resources Division, Korea Institute of Geosciences and Mineral Resources (KIGAM), 124 Gwahagno, Gajeong-dong, Yuseong-gu, Daejeon 34132, Korea; thenepallit@rediffmail.com (T.T.); hongha@kigam.re.kr (H.H.T.V.); ahnjw@kigam.re.kr (J.W.A.)

[4] Department of Bio-based Materials, School of Agriculture and Life Science, Chungnam National University, Daejeon 34132, Korea; chilakala_ramakrishna@rediffmail.com

***** Correspondence: kooltz77@kigam.re.kr

**Abstract:** Coal-derived power comprises over 39% of the world's power production. Therefore, a mass volume of coal combustion byproducts are generated and shifted the extra burden onto the economy and environment. Circulating fluidized bed combustion (CFBC) has been found to be a clean and ultimate technology for Korea's coal-fired power plants to have effective power generation from low-grade imported coal with reduced emissions. Efforts have been made to broaden the utilization of CFBC coal ash, and to promote sustainable development of CFBC technology. Investigations provided numerous evidences for coal ash to be a potential deposit for rare earths reclamation. However, the basic characteristics and the methods of rare earth mining from the CFBC bottom ash lack detailed understanding and are poorly reported. This study highlighted an insight of the CBFC bottom ash with respect to REEs concentration. Moreover, agents were tested as a means for leaching REEs from Samcheok CFBC bottom ash. The leaching tests were performed in relation to variations in concentration, time and temperature. The results were applied to identify suitable processes to leach REEs from the ash and clarify the potential valuation of CFBC bottom ash. The leaching conditions attained by ANOVA analysis for hydrochloric concentration, temperature, and time of 2 mol L$^{-1}$, 80 °C, and 12 h, were found to provide a maximum extraction of yttrium, neodymium and dysprosium of 62.1%, 55.5% and 65.2%, respectively.

**Keywords:** coal ash; circulating fluidized bed combustion (CFBC); rare earth elements (REEs); leaching

## 1. Introduction

Rare earth elements (REEs) comprise 15 elements, starting from the atomic number 57 (La) to 71 (Lu), of the periodic table encompassing scandium and yttrium. Some of them have been identified as critical and strategic materials [1–4]. The applications of REEs include catalysts for automobiles, batteries for hybrid cars, and permanent magnets for wind turbine [5,6]. REEs are indispensable and irreplaceable in emerging clean technologies due to their unique physicochemical properties [1,7]. The shift towards clean efficient technologies consequently increases REEs demand [3,8], rising from 33.9 to 51.9 kiloton of REEs during the 2016–2030 period [9]. Since 2009, the world market

has witnessed an exponential price rise of REEs due to the export restriction of rare earths from China [10,11], the largest REE supplier [12]. Thereby, increasing demand and the supplying risk have triggered global competition for REE resources and ticked upward request for REEs production, and consequently, an effort to ensure a stable supply stream in the long terms was initiated which involves mining and extraction from secondary sources [13].

However, the benefits of REEs recovery from waste products make it more attractive than primary mining. The recovery of REEs from conventional ores is a complicated multi-step process that requires extensive capital investments, consumes energy, generates massive undesired tailings, and has unintended environmental consequences [14,15]. The monopoly position and potential restrictions of China also boosted the rare earth industry into looking for new sources [16,17]. Rare earth recovery from industrial waste, end-used REEs bearing products, and mining residues (bauxite residue, coal tailings) has been considered as potential alternative sources of REEs [18,19]. There are numerous studies on the extraction of REEs from secondary sources [2,20]. REEs mining from red mud with extraction yields of 70–80% were documented [21–23]. There also exist studies on extraction and recovery of REEs from permanent magnets [24,25], or from fluorescent lamps [26].

The occurrence of rare earths in coal [6,27] and coal ash [28,29] have been understood for decades. Pan et al. (2019) [30] addressed coal ash as the most promising alternative source for REEs. The concentration of REEs in fly ash typically ranges from 200 to 550 ppm [31], and the value of 1400 ppm was reported by Mardon and Hower (2004) [32]. REE content in several particular coal fly ashes were proved to be equal to those in conventional ores [33]. The enrichment of REEs in coal ashes through a combustion process by double [29] to ten-fold [13,15] versus feed coals have been documented. Hence, coal combustion wastes offer possible resources for rare earths recovery [34]. The lower energy requirements with a net reduction of about 75% on $CO_2$ emissions than conventional mining per unit weight of REEs, are some additional advantages [15].

Globally, coal-derived power accounts for the major share of electricity generation by fuel, generating around 39% of global electricity [35] and supplying 41% of global demands. Korea was the sixth-largest consumer of coal worldwide in 2017 [36]. Coal-derived power remains the primary generation source and contributes more than one-third (39.6% in 2016) to Korea's power structure [37]. The advent of Circulating fluidized bed combustion (CFBC) technology facilitates using various coals including calorific coal as biomass fuel for generating higher net power when compared to pulverize combustion (PC) units [38], with reduced emissions, and operation and maintenance cost. Circulating fluidized bed (CFB) boilers are considered a clean and ultimate solution for Korea's coal-fired power plants, which heavily depend on cheap imported coal [39,40]. There are three CFBC power plants in operation in Korea [41]. However, considerable quantities of coal ash have been generated, amounting to 8.26 million tons in 2014 [42]. In South Korea, only 70% of the coal ash is recycled from total coal ash generation and most can be used in construction [42] remaining coal ash end up in ash ponds and landfills. Rare earths extraction from those CFBC power plants' ash expand REEs sources and the use of coal ash, and facilitate the development of CFBC power plants in terms of sustainability. Therefore, it is worthwhile proposing suitable techniques to recover rare earths from Korea's existing CFBC coal ash deposits.

Many research organizations are currently evaluating processes to recover rare earths from coal combustion wastes [43–45]. Several techniques for extraction and separation of individual REEs from waste coal ash have been summarized including physical [46] and chemical processes [47,48]. These processes generally start with the dissolution of rare earths using acid as the leaching agent, followed by separation of unwanted minerals from the leachate by filtration, and precipitation through solvent extraction, then recovery of REEs/REOs by hydro metallurgy. However, the unique chemical similarity of REEs causes the recovery process of REEs to be more complex.

In Korea, there are limited resources of rare earths and at the same time having landfill (coal ash residues, municipal solid waste residues, etc.) problems. To resolve these problems, efficient recycling technologies are required and adopted. A considerable recovery of critical rare earths such as

yttrium, neodymium and dysprosium can be achieved through a systematic combination of leaching and solvent extraction. The Nd/Dy significantly enhances the magnetic and optical properties by the addition of small amounts to other materials. The overall rare earth recovery process from coal ash samples is depicted in Figure 1.

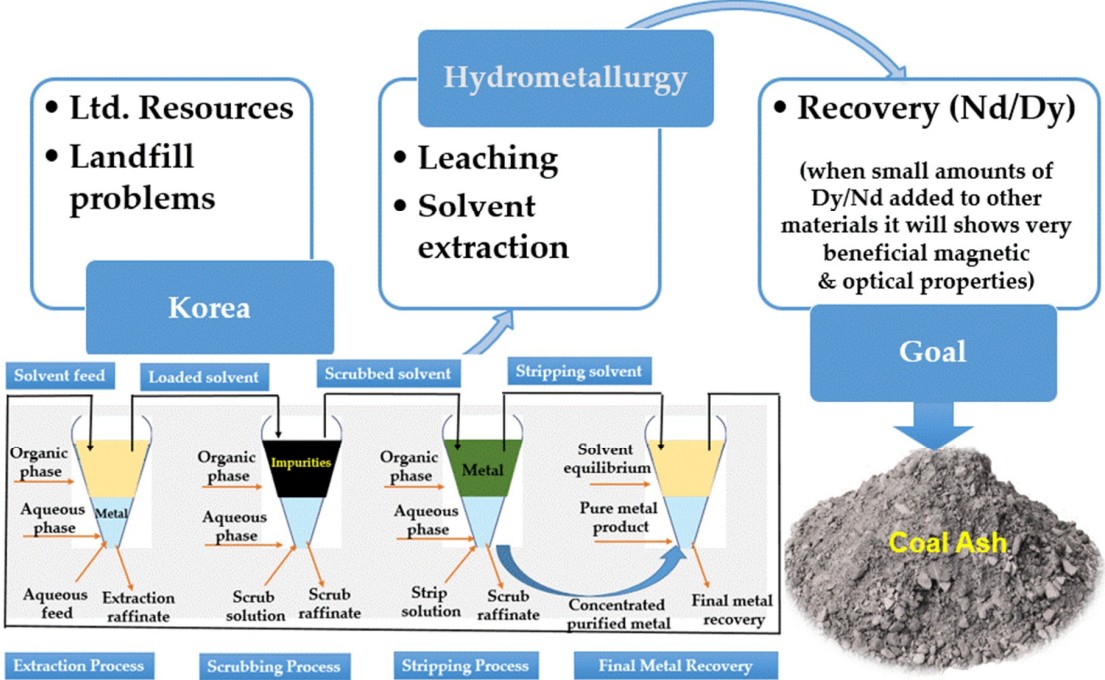

**Figure 1.** The overview of the rare earths recovery process from coal ash samples.

This study investigated the acid leaching methods applied on Samcheok CFBC bottom ash with a pulp density of 100 g $L^{-1}$ under different conditions that targeted to enhance REE (focus on Y, Nd, and Dy) leaching efficiency and reduce the amount of reagent used. The variables included the reagent type, reagent concentration, residence time, and temperature.

## 2. Materials and Methods

### 2.1. Materials

The bottom ash samples were collected from four coal power plants in South Korea namely Samcheok (Kospo), Yeosu (Hanwha), Shin Seocheon (Komipo), and Taean (Kepco). The Samcheok and Yeosu power plants utilize CFBC technology while Shin Seocheon and Taean power plants utilize pulverized combustion technology. Sulfuric acid (98%), hydrochloric acid (36%), and nitric acid (70%) were provided by Junsei chemical, Japan. 200 mL Erlenmeyer flasks (Duran) were used for REEs dissolution. Incubated shaker (IST-4075) was supplied by Jeiotech, Korea.

### 2.2. Methods

A mixture of 10 g of bottom ash in 100 mL of HCl were placed in an incubated shaker at a fixed speed of 200 rpm. After shaking for 12 h, the leachates were separated from the solid particles by filtration and stored in 50 mL Falcon tubes for Inductively Coupled Plasma-Mass Spectroscopy (ICP-MS) analysis. Based on value and significance, priority was given to the leaching of yttrium, neodymium, and dysprosium from the Samcheok bottom ash. Three leaching reagents such as sulfuric acid, hydrochloric acid, and nitric acid, were selected to investigate the influence of each agent on the leaching capacity. Based on the results from the screening tests the selected reagent was further used to explore the effects of concentration, time, and temperature. The optimum leaching conditions of

targeted rare earths from Samcheok CFBC bottom ash for HCl concentration, time and temperature were 2 mol $L^{-1}$, 12 h, and 80 °C respectively.

*2.3. Characterization*

The chemical composition of all bottom ash samples was determined by an X-ray fluorescence analyzer (Shimadzu, Japan). An X-ray diffractometer (XRD, D/MAX 2200, Rigaku, Japan) was used to assess the mineralogy in the Samcheok and Shin Seocheon samples. The concentration of individual REEs in four collected samples including Samcheok (0.6–2.36 mm), Yeosu (0.6–1.18 mm), Shin Seocheon (4.75–9.50 mm), and Taean (full size) was obtained via Inductively Coupled Plasma- Mass Spectroscopy (ICP-MS, Perkin Elmer DRC-II quadrupole).

## 3. Results and Discussion

The results of X-Ray Fluorescence (XRF) analysis for the four collected samples are detailed in Table 1. The calcium compounds presented in the two CFBC bottom ash samples were much higher than those from two Pulverized Combustion (PC) bottom ash samples. This variation was due to the limestone inserted to promote desulfurization in the CFB combustion chamber. The PC bottom ash, silicate, and aluminum-based compounds were dominant.

**Table 1.** X-ray fluorescence (XRF) analysis of coal bottom ash samples from different coal power plants in Korea.

| Composition (wt %) | Samcheok | Yeosu | Shin Seocheon | Taean |
|:---:|:---:|:---:|:---:|:---:|
| CaO | 44.04 | 33.45 | 1.08 | 4.29 |
| $SiO_2$ | 22.34 | 41.84 | 53.77 | 57.84 |
| $Fe_2O_3$ | 8.51 | 3.85 | 6.49 | 9.11 |
| $Al_2O_3$ | 6.49 | 4.61 | 30.08 | 20.77 |
| MgO | 6.23 | 2.69 | 0.9 | 1.32 |
| $K_2O$ | 0.91 | 0.62 | 3.74 | 1.08 |
| $Na_2O$ | 0.6 | 1.12 | 0.36 | 0.92 |
| $TiO_2$ | 0.31 | 0.23 | 1.83 | 1.13 |
| MnO | 0.12 | 0.08 | 0.06 | 0.1 |
| $P_2O_5$ | 0.05 | 0.07 | 0.18 | 0.31 |
| Ig loss | 2.74 | 3.77 | 1.49 | 3.44 |

The mineral compositions of Samcheok sample presented in Table 2a and Shin Seocheon samples mineral composition in Table 2b.

**Table 2.** (**a**). Mineral Compositions of Samcheok Coal Ash Samples by X-Ray Diffraction (XRD). (**b**). Mineral Compositions of Shin Seocheon Coal Ash Samples by X-Ray Diffraction (XRD).

| (a) | |
|:---:|:---:|
| **Sample Name** | **Composition (wt %)** |
| Samcheok samples | Anhydrite ($CaSO_4$), oldhamite (CaS), portlandite ($Ca(OH)_2$), Calcium silicate hydrate (C-S-H), quartz ($SiO_2$), lime (CaO), periclase(MgO), and aluminum trioxide ($Al_2O_3$) |

| (b) | |
|:---:|:---:|
| **Sample Name** | **Composition (wt %)** |
| Shin Seocheon samples | Mullite ($Al_6Si_2O_{13}$), quartz ($SiO_2$), sillimanite ($Al_2SiO_5$), portlandite ($Ca(OH)_2$), Calcium aluminum trioxide ($CaAl_2O_3$) |

The leach residue (solid particles) were heated at 80 °C for 12 h to observe the effect of leaching process on mineral compositions. Table 3 shows the residue's mineral phases of rare earths leaching test on Samcheok bottom ash utilized 2 mol $L^{-1}$ HCl at 25 °C for 24 h. Simpler mineralogy than the raw

bottom ash sample, the absence of portlandite, lime, periclase, and aluminum oxide and the formation of calcite, was observed.

**Table 3.** Mineral Compositions of Samcheok leaching residue when applied 2 mol L$^{-1}$ HCl as the leaching agent at 25 °C for 24 h.

| Sample Name | Composition (wt %) |
|---|---|
| Samcheok leaching residue | Anhydrite (CaSO$_4$), Calcite (CaCO$_3$), and quartz (SiO$_2$) |

The rare earths elemental composition is given in Table 4. The total concentration of REEs in Samcheok and Yeosu CFBC power plants were 78.4 mg/kg and 63.7 mg/kg, respectively. Those concentrations in Shin Seocheon and Taean PC power plants were approximately five times higher with 346.3 mg/kg and 371.8 mg/kg, respectively. La was the most abundant REE, followed by Nd and Y in Shin Seocheon sample. While in three other samples, the concentration of Ce was the largest, followed by Y, La, and Nd.

**Table 4.** Rare earths characterization of coal bottom ash samples from different coal power plants in (mg/kg).

| Element | Samcheok | Yeosu | Shin Seocheon | Taean |
|---|---|---|---|---|
| Y | 13.6 | 9.22 | 59.7 | 60.5 |
| La | 11.4 | 9.64 | 86.5 | 55.6 |
| Ce | 21.7 | 20.1 | 16.12 | 115 |
| Pr | 2.5 | 2.34 | 18.5 | 13.3 |
| Nd | 10.9 | 8.82 | 63.2 | 50.1 |
| Sm | 1.69 | 1.68 | 12.6 | 10.6 |
| Eu | 0.47 | 0.46 | 2.5 | 2.3 |
| Gd | 2.11 | 1.8 | 12.8 | 11 |
| Tb | 0.36 | 0.26 | 2.06 | 1.8 |
| Dy | 2.07 | 1.52 | 10.6 | 9.94 |
| Ho | 0.43 | 0.32 | 2.18 | 2.18 |
| Er | 1.22 | 0.86 | 5.72 | 6.24 |
| Tm | <0.20 | 0.2 | 0.86 | 0.94 |
| Yb | 1.28 | 0.8 | 5.38 | 5.88 |
| Lu | 0.59 | 0.2 | 0.82 | 0.94 |
| Th | 5.79 | 3.78 | 36.7 | 19.5 |
| U | 2.09 | 1.78 | 10.1 | 6.02 |
| total | 78.4 | 63.78 | 346.34 | 371.84 |

The experimental conditions were agent concentration, temperature, time, and pulp density of 3 mol L$^{-1}$, 25 °C, 24 h, and 100 g L$^{-1}$, respectively. We used three kinds of reagents for the leaching process and also for comparison studies. The three acids almost had a similar effect on the leaching process. Among those, HNO$_3$ is corrosive. The remaining acids have a similar price in Korea.

Leaching experiments for yttrium, neodymium, and dysprosium from Samcheok CFBC bottom ash using hydrochloric acid were performed with the variance of concentration, time, and temperature. The concentrations of the agent were in the range of 1, 2, and 3 mol L$^{-1}$. Firstly, 10 g of sample was added into an Erlenmeyer flask which contained 100 mL selected acid at a molar concentration of 1 mol L$^{-1}$. The mixture was then placed in an incubated shaker IST-4075 (Jeiotech, Korea) at a fixed speed of 200 rpm. The levels of temperature were set at 25, 50 and 80 °C. The residence time of slurry samples in the shaker were 3, 6, 12, and 24 h, respectively. The liquid solution was separated from the solid particles using filter paper (5C-110 mm, Advantec). A similar procedure was repeated for the leaching experiment of REEs with other selected molar concentration of the reagent. In each experiment, the shaking speed was set at 200 rpm, and the pulp density was fixed at 100 g L$^{-1}$. The concentration of REEs in leachate was characterized via ICP-MS analysis. The experimental conditions and corresponded results are shown in Table 5.

**Table 5.** Leaching test design and results.

| Test # | Test Conditions | | | % REE Extraction | | |
|---|---|---|---|---|---|---|
| | Acid Conc. (mol L$^{-1}$) | Temp (°C) | Time (h) | Y (%) | Nd (%) | Dy (%) |
| 1 | 1 | 25 | 24 | 35.8 | 29.5 | 35.7 |
| 2 | 2 | 25 | 24 | 38.2 | 31.7 | 39.1 |
| 3 | 3 | 25 | 24 | 36.0 | 33.0 | 36.2 |
| 4 | 2 | 25 | 3 | 36.5 | 32.2 | 38.1 |
| 5 | 2 | 25 | 6 | 35.5 | 31.7 | 36.2 |
| 6 | 2 | 25 | 12 | 59.4 | 48.2 | 60.3 |
| 7 | 2 | 50 | 3 | 33.0 | 31.1 | 35.2 |
| 8 | 2 | 50 | 6 | 38.0 | 36.6 | 41.0 |
| 9 | 2 | 50 | 12 | 38.3 | 37.2 | 39.1 |
| 10 | 2 | 80 | 3 | 42.0 | 40.8 | 44.9 |
| 11 | 2 | 80 | 6 | 48.4 | 45.9 | 50.7 |
| 12 | 2 | 80 | 12 | 62.1 | 55.5 | 65.2 |

## 3.1. Effect of Temperature

The effect of parameters on the leaching of metals was investigated and summarized [49]. Raising the temperature is believed to enhance the solubility of metals in the leaching process. A series of tests were performed to investigate the effect of temperature on the leaching of yttrium, neodymium, and dysprosium from Samcheok CFBC bottom ash. In these tests, the acid concentration, shaking speed, and pulp density was fixed at 2 mol L$^{-1}$, 200 rpm, and 100 g L$^{-1}$, respectively. The temperature was varied in the range of 25–80 °C. The leaching experiments were conducted for 3 and 12 h for each of Y, Nd, and Dy. Figure 2 shows the effect of temperature on the REE extraction. When the residence time was 3 h, the leach rare earths first decreased and then increased with increasing temperature. At a temperature of 80 °C, the concentration of rare earths in the leachate reached its maximum. The similar trend was also observed in 12 h leaching tests.

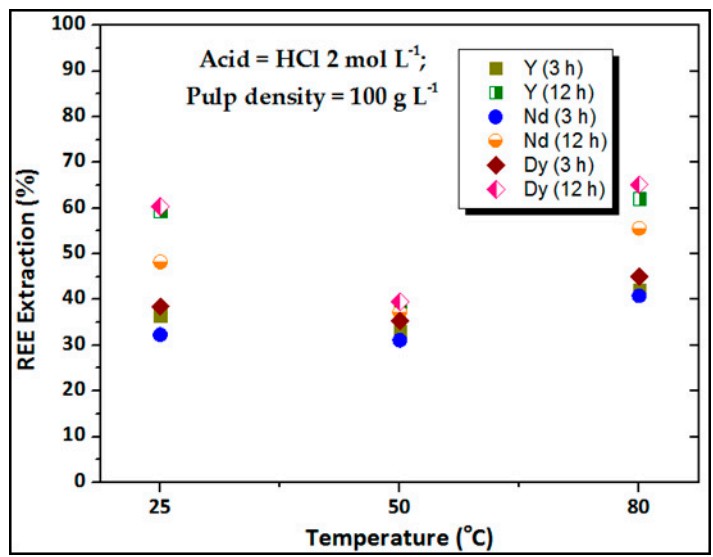

**Figure 2.** Effect of temperature on extraction of Y, Nd, and Dy.

## 3.2. Effect of Acid Concentration

The reagent concentration was reported to have a significant effect on the dissolution of metals in the leaching process [49]. The leaching rate rose linearly as the concentration raising until it reached the peak. After this point, the increase in the agent concentration did not provide a higher leaching rate, in addition, causes a slight reduction. The influence of agent concentration on the

leaching rate of yttrium, neodymium, and dysprosium was checked by varying the concentration of hydrochloric acid in the experiments. Hydrochloric acid at the concentrations of 1, 2 and 3 mol $L^{-1}$ was sequentially adopted while the temperature and time were fixed at 25 °C and 3 h, respectively. As shown in Figure 3, the concentration of yttrium and dysprosium in the leach solution increased with the increase of hydrochloric concentration from 1 to 2 mol $L^{-1}$ and then decreased when the hydrochloric concentration was 3 mol $L^{-1}$.

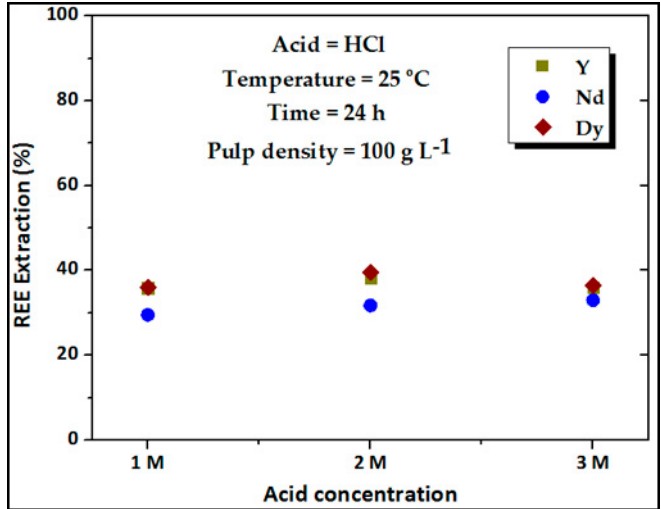

**Figure 3.** Effect of acid concentration on the extraction of Y, Nd, and Dy.

### 3.3. Effect of Leaching Time

In order to explore the effect of the residence time of leaching test on the leaching of yttrium, neodymium, and dysprosium by utilizing hydrochloric acid 2 mol $L^{-1}$, the residence time was varied from 3 to 24 h. At the temperature of 25 °C, the leach rare earths initially increased (Figure 4) and attained a peak after 12 h. But it followed by the reduction of leach rare earths as the residence time for leaching continually increased up to 24 h.

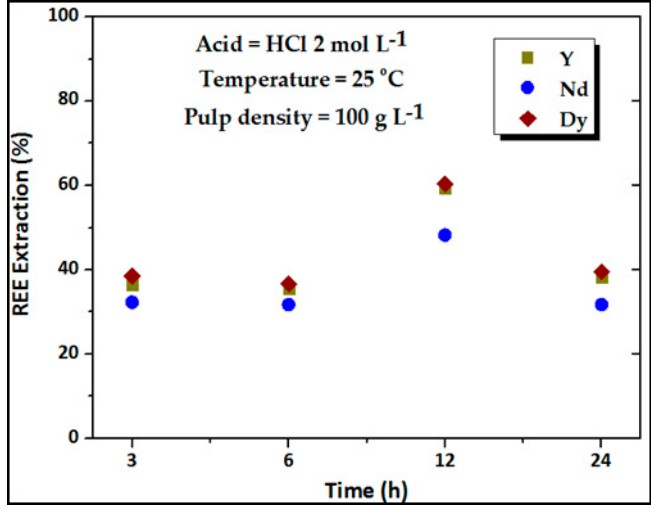

**Figure 4.** Effect of time (residence time) on the extraction of Y, Nd, and Dy.

### 3.4. Estimation of Percentage (%) of REE Extraction at Optimal Conditions

Preliminary results were input to Minitab software to explore the relationships among leaching variables (concentration, time, temperature) and leaching efficiency of yttrium, neodymium,

and dysprosium. ANOVA analysis of main effects plot to was outlined in Figure 5. The trends of rare earths content in leach solution due to varying the variables' value (Figure 5) are similar to the scatter plot of rare earths leach with respect to variation of temperature (Figure 2), hydrochloric concentration (Figure 3), and residence time (Figure 4). The respective values of the concentration, time, and temperature of 2 mol L$^{-1}$, 12 h, and 80 °C are proposed to be optimal conditions for leaching targeted rare earths from Samcheok CFBC bottom ash.

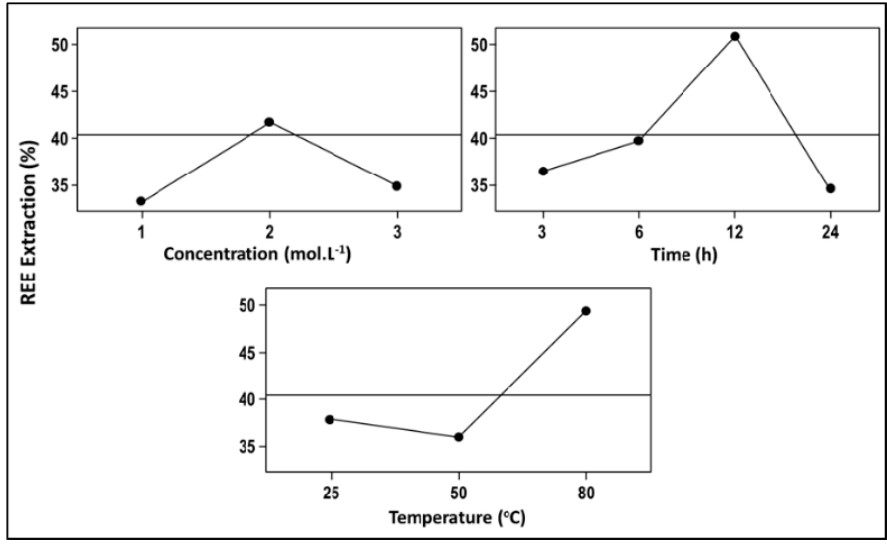

**Figure 5.** Leaching variables to rare earths extraction.

## 4. Conclusions

This research represents one of the first extended studies reporting rare earths content and their extractable fraction from bottom ash generated at CFBC coal-fired power plants in Korea. Characterization results of CFBC bottom ash showed wide variety of rare earths but at a lower quantity compared to those generated from utilities using PC technology. Present work focuses on extraction of critical rare earths (Y, Nd, and Dy) by acid leaching with variations in concentration, residence time, and temperature. The conditions of hydrochloric acid 2 mol L$^{-1}$ for 12 h leaching time at 80 °C predicted by ANOVA are expected to provide maximum leaching extraction. However, extraction of yttrium, neodymium and dysprosium were 62.1, 55.5 and 65.2%, respectively, at optimal conditions. There is an utmost requirement for improved methods of leaching and extraction of rare earths. In this respect, the economic feasibility and environmental sustainability of CBFC bottom ash as an alternative rare earths resource will hinge upon future development in scalable extraction technologies.

The simple mineral-acid leaching of CFBC material by this technique was significant. Pulverized material likely is more promising since the REEs are more concentrated since no secondary calcium sulphate from limestone scavenging is present.

The future prospects of this research are to find out the solutions for critical rare earth supply and demand. South Korea is one of the leading countries in the manufacturing of electronic devices and shows great potential to establish an electronics hub center. But, Korea has very limited natural resources and depends on recycling technology. Currently, Yttrium, Neodymium, Dysprosium are highly demanded in clean energy, electronic devices applications and are relatively very expensive. Our goal and motto is the development of optimum conditions for 100% recovery of rare earth from industrial by-products such as coal ash and municipal solid wastes by leaching. Now we are an attempt for CFBC type bottom ashes, but in the future, we are interested in the other kinds of low-grade coals and coal ash samples which are available in South Korea.

**Author Contributions:** L.Q.T., T.T., R.C. did the experiments and analyzed the data and wrote the manuscript. K.J.—supervision. H.H.T.V. supported the analysis of the samples. A.J.W. corrected the final manuscript and agreed to submit this data to sustainability journal.

**Funding:** This research was funded by Energy Technology development Project of the Korea Institute of Energy Technology Evaluation and Planning funded by Ministry of Trade, Industry, and Energy (MTIE). Grant Number [20141010101880].

**Conflicts of Interest:** The authors declare no conflict of interest.

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
