# Peer review of "Leaching Characteristics of Low Concentration Rare Earth Elements in Korean (Samcheok) CFBC Bottom Ash Samples"

_sustainability, doi:10.3390/su11092562_

Reviewer 1 Report

It was very interesting and informative with basic research methodologies. The XRF and XRD and extraction studies indicate the significance of the study in this area. It can be accepted after minor revisions of the following comments. 

-The resolution of all figures should be improved.

-The references are too much can be reduced with recent one or reviews.

"If you included these following suggestions, the quality of paper should be improved further...

-In fig. 8 X-axis caption should be added.

-what are the future prospects of this research? Those details should include.

-The schematic representation of leaching methodology with optimum conditions should include."

Author Response

Answers to the Reviewers comments to the article Sustainability- 450231 - 1st Reviewer

General Comments

It was very interesting and informative with basic research methodologies. The XRF and XRD and extraction studies indicate the significance of the study in this area. It can be accepted after minor revisions of the following comments.

Ans) Thank you very much for your valuable corrections and suggestions to improve the manuscript quality. We are greatly thankful to you.

Specific comments

1. The resolution of all figures should be improved.

Ans) In the revised manuscript, we modified the graphs resolution

2. The references are too much can be reduced with recent one or reviews?

Ans) Yes sir. We removed some of the references

3. "If you included these following suggestions, the quality of paper should be improved further...

Ans) thank you sir, we followed your suggestions

4. In fig. 8 X-axis caption should be added

Ans) Yes. We modified the fig. 8

5. What are the future prospects of this research? Those details should include.

Ans) Yes we added in the revised manuscript

6. The schematic representation of leaching methodology with optimum conditions should include

Ans) Yes we added the optimum leaching conditions in the methodology section

Reviewer 2 Report

The paper need substantial reconsideration ands language editing.

The leaching efficiency, or metal extraction in % units is the most appropriate metric of a leaching operation - not the concentration. All Figures need to be redone. Embedded comments are provided in the attached manuscript.

Author Response

Answers to the 2nd Reviewers comments to the article Sustainability- 450231

Ans: Thank you very much for your valuable suggestions and comments.

1) In line numbers 85-87, not clear what excluded-English grammar problem. Please simplify and rephrase.

Ans) Yes. We modified the sentence in revised manuscript

2) In line no.102, not clear schematic at all. Bottom part is too small. Any schematic such as this must be discussed and explained in the text. Otherwise should be deleted.

Ans) Yes. We included the discussion about the figure 1 in the revised manuscript from line no. 93-98.

3) Line no.114, were they agitated? How

Ans) Yes. We used incubated shaker (IST-4075) for agitation and added this information in the revised manuscript line no.112.

4) Line no.115, explain how in the methods section below. filtered? washed?

Ans) Yes. We added this information in the methods section from line no.114-116 in the revised manuscript.

5) From Line no.141-146, Figures 2a and 2b are not informative. The authors better list the mineralogy in Table form similarly to Table 1.

Ans) Yes. We listed minerology in a separate tables 2 (a) and 2 (b) and table 3 in line no’s. 139-142 and 148 respectively in the revised manuscript.

6) Line no.152, Similarly to Figures 2a and 2b, delete and just report the mineral phases.

Ans) Yes. We listed minerology in a separate table 3 in line no. 148 in the revised manuscript.

7) Suggest to give numbers with one decimal place (Line no.160)

Ans) Yes. We added the decimals with one place only in the line no.s 151 and 153 in the revised manuscript.

8) Efficiency is the wrong word. Do you mean % extraction? (Line no.169), Not important numbers - Only the % extraction is important. Should convert. The same for ALL subsequent figures and tables.

Ans) Yes. We added the % instead of efficiency and clearly mentioned in the Table 5 (a) and Table 5 (b) in the revised manuscript in the line no’s 177 and 178.

9) if extraction is was only 3.5 %., the results are not worthy of publication at all (line No.174).

Ans) Yes. We thoroughly investigated and found our mistake and rectified. Previously, we forgot the dilution factor and dilution of 10g ash/100 ml (10% dilution) without multiply with 10, we gave those values.

Now we included the diluted factor (multiply with 10).  In the present tables we calculated correctly, now the extraction percentage was more than 60% and clearly showed in the Table 5 (b), line no’s 178 in the revised manuscript.

10) Not very meaningful given the low extractions and the too few tests made. (Line no.229).

Ans) Yes sir. We are doing continuously on this research work. Up to this paper writing, we did the preliminary investigation of coal ash samples particularly low concentrated samples. Kindly consider this manuscript. Based on these results we are trying to recovery by using solvent extraction method by applying suitable extractants for low concentrated rare earths in the coal ash samples.

11) About English grammar

Ans) we checked English grammar throughout the manuscript. We are highly thankful to your valuable suggestions, recommendations and grammar corrections.

Reviewer 3 Report

Interesting study!

A couple of comments:

) Graphs labeled "Leaching Efficiency" don't really show efficiency, which would typically be a ratio. They really show "Recovery" or "Amount Leached."

) Figure 4 also says "Leaching Efficiency" but I have no idea what is meant. 3.39? Also, the graph is misleading because the bars don't go to zero. So the difference between, say, sulfuric and hydrochloric, 3.39 vs 3.48, is only .09/3.39 = about 2.6 % That's probably within the error limits of the analyses. So really no significant differences between these acids.

) It would be good to present the real efficiency of the process, that is, how much of the REE content of the starting material was leached. My calculation*, if correct, indicates that a typical recovery of the REEs in most of these experiments was around 15%.  That would lead to the conclusion that perhaps this process is not promising for the CFBC material.

) That disheartening conclusion is important since it suggests that simple mineral-acid leaching of CFBC material by this technique is probably a dead end economically. Pulverised material likely is more promising since the REEs are more concentrated since no secondary calcium sulfate from limestone S-scavenging is present.

* 78.4 ppm = 78.4 micro g / g   in a 10 g solid = 784 micro g. 

say 1000 micro g / L (Fig. 8) typical recovery    in 100 mL = 100 micro g.

100 micro g recovery / 784 micro g start  =  about 15%

Author Response

Answers to the Reviewers comments to the article Sustainability- 450231 – 2nd Reviewer

General Comments

Interesting study!.

Ans) Thank you very much for your comment on the manuscript. Thank you for your valuable corrections and suggestions to improve the manuscript quality. We are greatly thankful to you.

A couple of comments:

1. ) Graphs labeled "Leaching Efficiency" don't really show efficiency, which would typically be a ratio. They really show "Recovery" or "Amount Leached.".

Ans) Yes. We removed the leaching efficiency word and added “amount leached” in the graphs in revised manuscript

2. Figure 4 also says "Leaching Efficiency" but I have no idea what is meant. 3.39? Also, the graph is misleading because the bars don't go to zero. So the difference between, say, sulfuric and hydrochloric, 3.39 vs 3.48, is only .09/3.39 = about 2.6 % That's probably within the error limits of the analyses. So really no significant differences between these acids.

Ans) Yes sir. We modified the graph.  The three acids are almost having similar effect on leaching process. For comparison studies we used three reagents. But, HNO3 is corrosive. The remaining acids are similar price in Korea. The HCl reagent availability is sooner than H2SO4 during the experiments. Still we are continue these leaching process with different coal ash samples and with different reagents.

3. ) It would be good to present the real efficiency of the process, that is, how much of the REE content of the starting material was leached. My calculation*, if correct, indicates that a typical recovery of the REEs in most of these experiments was around 15%.  That would lead to the conclusion that perhaps this process is not promising for the CFBC material.

Ans) We will follow your suggestions.

4. That disheartening conclusion is important since it suggests that simple mineral-acid leaching of CFBC material by this technique is probably a dead end economically. Pulverised material likely is more promising since the REEs are more concentrated since no secondary calcium sulfate from limestone S-scavenging is present.

* 78.4 ppm = 78.4 micro g / g   in a 10 g solid = 784 micro g.

say 1000 micro g / L (Fig. 8) typical recovery    in 100 mL = 100 micro g.

100 micro g recovery / 784 micro g start  =  about 15%

Ans) Thank you sir. We added in the final conclusion part.

Round  2

Reviewer 2 Report

The revised manuscript barely makes the paper suitable for publication. There were many language issues and inconsistencies in the revised manuscript.  Also, it does not make sense for 7 authors to publish preliminary results, which implies that the authors admit upfront that results may change in the future. A different title is recommended.  

Again, embedded comments are provided in the manuscript for the authors' note.

Author Response

Answers to the 2nd Reviewer -2nd round comments to the article Sustainability- 450231

General comments

The revised manuscript barely makes the paper suitable for publication.

Ans: Thank you very much once again for your valuable suggestions and comments.

There were many language issues and inconsistencies in the revised manuscript.

Ans: Yes sir. We checked thoroughly and modified.

Also, it does not make sense for 7 authors to publish preliminary results, which implies that the authors admit upfront that results may change in the future.

Ans: The final (6, &7) are Center Director(Ji Whan Ahn, guest editor of this special issue) and Corresponding author, the remaining 4 authors all are did experiments, samples collection from other city, leaching experiments, analysis, writing and editing everything. That’s what we added.

A different title is recommended. 

Ans: Yes sir. We modified the title and included in the revised manuscript.

Specific Comments

1) In line number 2, delete “Preliminary investigation of” and add concentration instead of concentrated in the title

Ans) Yes. We deleted the Preliminary investigation of and concentrated in the title in revised manuscript in line no.2-4.

2) In line no.118-119, deleted “after leaching”.

Ans) Yes. We deleted the words “after leaching” in the revised manuscript in line no. 116.

3) Line no.126, incorrect symbol of degree

Ans) Yes. We modified and added the correct version of temperature units in the revised manuscript line no.123 and also throughout the manuscript.

4) Line no.164, add have instead of “are“.

Ans) Yes. We added have instead of are line no.164 in the revised manuscript.

5) Line no.165, removed from the text.  

Ans) Yes. We removed the sentence “The HCl reagent availability is sooner than H2SO4 during the experiments” in the revised manuscript.

6) From Line no.177~179, deleted the repeated tables, only the % extraction should be shown in these 3 columns.

Ans) Yes. We deleted the table 5 (b) in line no. 181 in the revised manuscript.

7) Line no.190, Wrong term.  % is not an amount. Use consistent terminology throughout. REE Extraction is the best. Replace leaching efficiency and all other terms.

Ans) Yes sir. We added the “REE Extraction” in all the tables and figures in the revised manuscript (line no.184).

8) Delete the amount in figure 2 caption, figure 3, figure 4.

Ans) Yes. We deleted the “amount” word in figure 2, figure 3, and figure 4 captions in the line no.188, 201 and 210.

9) Delete the word “recovery”, add extraction in the figure 5 caption (line 228).

Ans) Yes. We deleted the word “recovery” and added “extraction” in the figure 5 caption in the line no. 221.

10) Delete the word “leaching efficiency” and add extraction in the line no.237.

Ans) Yes. We deleted the word “leaching efficiency” and added “extraction” in the line no.230.

We thoroughly checked the grammar throughout the manuscript.

Reviewer 3 Report

Figure 5, 6, and 7 are still not labelled correctly. Both the y axis and the caption say "efficiency" instead of "recovery."

Fig. 5 y axis is mg/l  instead of micro g/l.

The abstract, text, and conclusions should state that the efficiency or percent recovery of the extraction was approximately 15%. The abstract does not say anything about this, and the conclusions merely state that the efficiency "is low." There is no reason not give the quantitative conclusion, ~15%, rather than a qualitative "low."

Author Response

Answers to the 3rd Reviewers comments to the article Sustainability- 450231

1) Figure 5, 6, and 7 are still not labelled correctly. Both the y axis and the caption say "efficiency" instead of "recovery."

Ans) Yes. We removed the leaching efficiency word and added “amount leached” in the graphs in revised manuscript

2) Fig. 5 y axis is mg/l  instead of micro g/l.

Ans) Yes. We modified the figure 5 in the revised manuscript

3) The abstract, text, and conclusions should state that the efficiency or percent recovery of the extraction was approximately 15%. The abstract does not say anything about this, and the conclusions merely state that the efficiency "is low." There is no reason not give the quantitative conclusion, ~15%, rather than a qualitative "low."

Ans) Yes. We added quantitative value of leaching efficiency in abstract, text, and conclusions in revised manuscript.
